# Optimal Model and Algorithm of Medical Materials Delivery Drone Routing Problem under Major Public Health Emergencies

**Lijing Du [1], Xiaohuan Li [1,\*], Yuan Gan [2] and Kaijun Leng [3]**

[1] School of Safety Science and Emergency Management, Wuhan University of Technology, Wuhan 430070, China; dulijing@whut.edu.cn
[2] School of Economics and Finance, Guizhou University of Commerce, Guiyang 550014, China; 201520198@gzcc.edu.cn
[3] Research Center of Hubei Logistics Development, Hubei University of Economics, Wuhan 430205, China; lengkaijun@hbue.edu.cn
[\*] Correspondence: lixh_whut@163.com

**Abstract:** To reduce distribution risk and improve the efficiency of medical materials delivery under major public health emergencies, this paper introduces a drone routing problem with time windows. A mixed-integer programming model is formulated considering contactless delivery, total travel time, and customer service time windows. Utilizing Dantzig–Wolfe decomposition, the proposed optimization model is converted into a path-based master problem and a pricing subproblem based on an elementary shortest path problem with resource constraints. We embed the pulse algorithm into a column generation framework to solve the proposed model. The effectiveness of the model and algorithm is verified by addressing different scales of Solomon datasets. A case study on COVID-19 illustrates the application of the proposed model and algorithm in practice. We also perform a sensitivity analysis on the drone capacity that may affect the total distribution time. The experimental results enrich the research related to vehicle routing problem models and algorithms under major public health emergencies and provide optimized relief distribution solutions for decision-makers of emergency logistics.

**Keywords:** major public health emergencies; drone routing problem; medical materials delivery

## 1. Introduction

Major public health emergencies have had a great influence on human health and social development. Examples in recent years include Ebola in 1976, HINI in 2009, and SARS in 2003. Coronavirus disease 2019 (COVID-19) is the most far-reaching global public health emergency. When an infectious disease occurs, medical supplies play an essential role in reducing dissemination risk, guaranteeing people's lives and health. With limited time and resources, emergency logistics decision-makers must make the best decisions regarding the allocation of limited time, funds, and other resources.

Nonetheless, the highly infectious and harmful characteristics of the virus have made the prevention of COVID-19 relatively difficult. Many areas adopt blockade policies to alleviate the spread of the epidemic [1]. Early and strict lockdown measures, rapid testing, and increased media campaigns are crucial to curb the pandemic [2]. The blockade policy also greatly limits the distribution of medical supplies. Hence, it is necessary to achieve timely distribution of medical supplies in the case of inaccessible transportation networks and highly contagious viruses.

With the continuous development of high technology, new models of delivery—mainly drones—have come into the limelight. Although the application of drones in logistics is in its infancy, commercial practice has already begun. SF began researching drone deliveries

as early as 2012, JD Logistics Laboratory also began drone testing in June 2016, and some villages around Suqian achieved delivery by drone during the "618" period in 2017. The rapid global spread of COVID-19 has resulted in interruptions to supply chains and people's lives [3], drones or unmanned aerial vehicles (UAVs) can contribute significantly to the fight against the COVID-19 pandemic [4]. Drones can be adopted to distribute viral tests to potentially infected patients [5], provide food delivery services [6], enable timely identification of infected people on the road, detect unmasked people [7], spray disinfection on the street, data analysis, delivery medical supplies, and make announcements [8]. Drones fly relatively fast, which can well ensure their work efficiency. They are very flexible, and not easily restricted by the terrain. Contactless delivery of drones can greatly reduce human contact and reduce the risk of delivery under epidemic situations. Given the above, this study will choose drones for medical materials delivery, which will contribute significantly to slowing down epidemic spread and ensuring people's health to a certain extent.

This work contributes in three ways. Firstly, this study focuses on solving the medical materials distribution problem during major public health emergencies, which is a research hotspot for the present. Secondly, we establish a model of drone routing problem under major public health emergencies, where drones can achieve contactless distribution and are less affected by the actual road network. Finally, we adopt a combination of the column generation (CG) and pulse algorithm (PA) to solve the model to ensure the quality and robustness of the solution.

The rest of the paper is organized as follows: Section 2 reviews the relevant literature. Section 3 represents problem description, establishes mixed-integer programming formulation, and makes modifications to the model. Section 4 gives the design of the CG and PA. Algorithm validation and a case study are shown in Section 5. Section 6 makes conclusions and points out future research.

## 2. Literature Review

### 2.1. Models of Medical Materials Delivery VRP under Major Public Health Emergencies

Several scholars related results on models of medical materials distribution vehicle routing problem (VRP) under major public health emergencies are shown in Table 1. These studies consider VRP under major public health emergencies, where the supplies to be transported are usually emergency supplies such as masks [9], vaccines [10], and food [11]. As seen in Table 1, travel cost, travel time, and infection risk are the three most frequently considered travel factors. With the availability of infectious disease supplies, it is important to consider the risk of infection. For this reason, some works included safety scores. Only a few papers focus on increasing resident satisfaction and minimizing overall violations, which are of equal importance in humanitarian logistics.

In an emergency logistics network, emergency decision-makers need to consider specific objectives to achieve optimal distribution of medical supplies. Dawei Chen increased residents' satisfaction with food allocation services in a closed gated community under COVID-19 [11]. Considering the open path vehicle routing problem in the production and distribution of face shields, Joaquín Pacheco concentrated on minimizing the time of the longest route [9]. Erfan Babaee Tirkolaee developed a novel mixed-integer linear programming model for strengthening the treatment of infectious medical waste [12]. Emre Eren targeted safety scores and total distance traveled [13]. Min-Xia Zhang worked on minimizing the sum exposure duration of all individuals [14], and Yuzhan Wu employed multiple autonomous ground vehicles for last-mile transportation to minimize total delivery costs [15]. Considering the massive fresh agri-product demand under COVID-19, Yiping Jiang constructed an optimization model based on average response time, the likelihood of infectious disease risk, and transportation resource utilization [16].

**Table 1.** Models of medical materials delivery VRP under major public health emergencies.

| Materials | Objectives (Min↓/Max↑) [1] | Constraints | Model | References |
|---|---|---|---|---|
| Food | Residents' satisfaction↑ | Contactless distribution, Homogeneous vehicle | Mathematical programming | Dawei Chen [11] |
| Face shields | Travel time↓ | Open path, Homogeneous vehicle | Mathematical programming | Joaquín Pacheco [9] |
| Materials | Travel time↓ | Time windows, Contactless delivery, Homogeneous vehicle | Mixed-integer linear program | Cheng Chen [17] |
| Medical waste | Travel time↓, Total violation↓, Infection risk↓ | Time windows, Fuzzy demand, Heterogeneous vehicle, | Mixed-integer linear program | Erfan Babaee Tirkolaee [12] |
| Medical waste | Travel distance↓, Safety scores↑ | Homogeneous vehicle | Linear program | Emre Eren [13] |
| High-risk individuals | Infection risk↓ | Heterogeneous vehicle | Mathematical programming | Min-Xia Zhang [14] |
| Essential materials | Travel cost↓, Travel time↓ | Time windows, Homogeneous vehicle | Bi-objective mixed-integer programming | Yong Wang [18] |
| Emergency resource | Travel time↓ | Contactless distribution, Heterogeneous vehicle | Mixed-integer linear program | WEI GAO [19] |
| Emergency materials | Travel cost↓ | Heterogeneous vehicle | Mixed-integer programming | YuzhanmWu [15] |
| Municipal solid waste | Travel cost↓, Infection risk↓ | Time windows, Split delivery, Heterogeneous vehicles | Mixed-integer linear program | Kannan Govindan [20] |
| Fresh agri-product | Travel cost↓, Infection risk↓, Resource utilization↓ | Split delivery | Mathematical programming | Yiping Jiang [16] |
| Vaccine | Total number of infected individuals↓, fixed cost of vehicles↓ | Heterogeneous vehicles, Priority groups | Mathematical programming | Nafseh Shamsi Gamchi [10] |

[1] "**Min↓/Max↑**" means the minimize and maximize objectives, respectively.

Vehicle capacity constraints, vehicle flow constraints, elimination of sub-paths, and variable constraints are the basic constraints of the VRP model. Almost every model may contain the above constraints [9–20]. It performed that there is little difference in the proportion of documents considering homogeneous vehicles and heterogeneous vehicles. For multiple types of vehicles, there are also interaction ends of different types of vehicles [12,14,15,19,20]. The customers required emergency supplies to be accepted within a specified time frame, thereby introducing time window constraints [12,17,18,20]. Multi-item packaging and split delivery of fresh produce in the context of a large-scale epidemic can increase the efficiency of relief delivery [16]. In most of the literature, customer demand was known in advance, but there are also documents that obscure customer demand [12].

According to the complexity and diversity of constraints and objectives, there will also be some differences in the established model. A range of articles [8,12,17,19,20] were modeled in the form of a mixed-integer linear programming formulation. Additionally, mathematical programming [9,11,14] and mixed-integer programming [15,18] were also usually used for modeling.

From the above literature, contactless delivery can reduce the risk of infection during the epidemic, so unmanned delivery will become a trend in the future. Especially in recent years, the emergence of drones and unmanned vehicles has provided an opportunity for contactless delivery. For strict traffic control, the utilization of drones for distribution will highlight great advantages. It will also be more realistic to consider the time window for the delivery of services. In terms of objective, minimizing the total travel time matches the characteristics of timeliness of emergency logistics.

### 2.2. Algorithms of Medical Materials Delivery VRP under Major Public Health Emergencies

As can be seen from Table 2, most researchers focus on two types of algorithms: heuristic algorithms and mathematical programming algorithms. Approximately 67% of the work use heuristic algorithms to solve VRP. The employed solution includes tabu search (TS) algorithm, bee colony algorithm, a two-stage hybrid heuristic algorithm, floyd algorithm, and particle swarm optimization (PSO) algorithm, genetic algorithm, and so on. Dawei Chen embedded the tabu search operator into an artificial bee colony algorithm to solve problems [11]. And the proposed algorithm was found to have better performance compared to various algorithms. WEI GAO utilized an iterative improvement algorithm to solve random instances, which improved the speed of the solution by more than 10% [19]. Min-Xia Zhang proposed a hybrid algorithm based on water wave optimization (WWO) metaheuristic and neighborhood search with a high solution rate [14]. Yiping Jiang designed an improved genetic algorithm based on solution features (IGA-SF) to address the integrated model with multiple decision variables [16]. Numerical results showed that the proposed IGA-SF had great superiority in terms of CPU running time and number of iterations for comparison with genetic algorithms. The above researchers utilize heuristic algorithms to solve related models. Of course, some researchers use mathematical programming methods. Erfan Babaee Tirkolaee solved sustainable multi-trip location-routing problems with time windows (MTLRP-TW), employing a fuzzy chance-constrained programming approach [12]. Emre Eren adopted the analytic hierarchy process (AHP) to obtain safety scores [13]. Kannan Govindan applied a fuzzy goal programming approach for solving the proposed bi-objective model and used data related to 13 nodes of medical waste production in western Tehran to evaluate the efficiency of the proposed model and solution method [20]. Data cases were based on real cases, standard data sets, and randomly generated data for algorithm performance testing. Nevertheless, due to the different objective functions and constraints, it is impossible to directly compare the quality of each algorithm, but the algorithm designed in the given article certainly shows good performance in the calculation examples.

**Table 2.** Algorithms of medical materials distribution VRP under major public health emergencies.

| Type | Algorithm | Instances | References |
|---|---|---|---|
| Heuristic algorithm | Two-stage metaheuristic algorithm | Solomon data test | Cheng Chen [17] |
| | Iterative improvement algorithm | Randomly instances | WEI GAO [19] |
| | Bee colony algorithm | Randomly generated | Dawei Chen [11] |
| | Floyd algorithm and PSO algorithm | Randomly generated | Yuzhan Wu [15] |
| | TS algorithm | Cartographic data for the province of Burgos | Joaquín Pacheco [9] |
| | WWO metaheuristic, neighborhood search | Seven real-world instances | Min-Xia Zhang [14] |
| | A two-stage hybrid heuristic algorithm | A real-world case | Yong Wang [18] |
| | An improved genetic algorithm | A real case study | Yiping Jiang [16] |
| Mathematical programming algorithm | A fuzzy chance-constrained programming approach | A real case study | Erfan Babaee Tirkolaee [12] |
| | AHP | A real case study | Emre Eren [13] |
| | Fuzzy goal programming approach | A real case study | Kannan Govindan [19] |
| | Dynamic programming | A real case study | Nafseh Shamsi Gamchi [10] |

From the existed references, the majority of works propose various heuristic algorithms and mathematical programming to solve routing problems that arise in practice, while few works adopt exact algorithms to solve problems. The heuristic algorithm shows good performance in the solution time, but it cannot guarantee the optimal solution. Thus, the paper will adopt a mixed CG and PA to solve the proposed problem. The proposed algorithm performs well in terms of guaranteeing the quality of the solution. The CG was proposed by Danzig in 1960 and it has been applied in recent years. Alain Chabrier solved the elementary shortest vehicle routing problem with the CG [21]. Zhi-Long Chen proposed a dynamic method based on the CG to solve a dynamic VRP with hard time windows [22]. Eunjeong Choia presented a tight integer programming model and successfully adopted the CG approach to a vehicle routing problem with time windows (VRPTW) [23]. Generally, the CG algorithm was also commonly used to address the lower bound of the branch-and-price approach [24]. Prahalad Venkateshan developed an efficient algorithm based on the CG for solving a pickup and delivery problem [25]. Leonardo Lozano firstly proposed an exact solution method capable of handling constrained shortest paths of large-scale networks in a reasonable time [26]. Then, the author extended the proposed algorithm and proposed a novel bounding scheme in 2016 [27].

## 3. Problem and Mathematical Model

### 3.1. Problem Description

The paper concentrates on the face shields distribution drone routing problem with time windows (DRPTW). Firstly, a homogeneous fleet of drones carrying a certain amount of face shields are dispatched from the distribution center to provide medical materials distribution services for many medical materials demand points. Consider a directed graph

$G = (N, E)$ provided in Figure 1, where $N = \{0, 1, 2, \ldots, n, n+1\}$ represents the set of all nodes and $E = \{(i, j) : i, j \in N, i \neq j\}$ denotes the set of all available links in the network. Set $N$ contains three subsets: $\{0\}$, $\{n+1\}$ and $C$. $\{0\}$ and $\{n+1\}$ represent the starting and the ending nodes, representing the distribution center. $C = \{1, 2, 3, \ldots, n\}$ is the set of community closed, below referred to as customers. Each customer $i$ has a known demand $q_i$ and each customer demand can be satisfied by a drone exactly once in required time window $[a_i, b_i]$. Deliveries outside the time window will not be tolerated. The service time of each customer $i$ will be considered according to the actual situation. Drones delivery services must begin and end their routes within a specified time window $[a_0, b_0]$ at the distribution center.

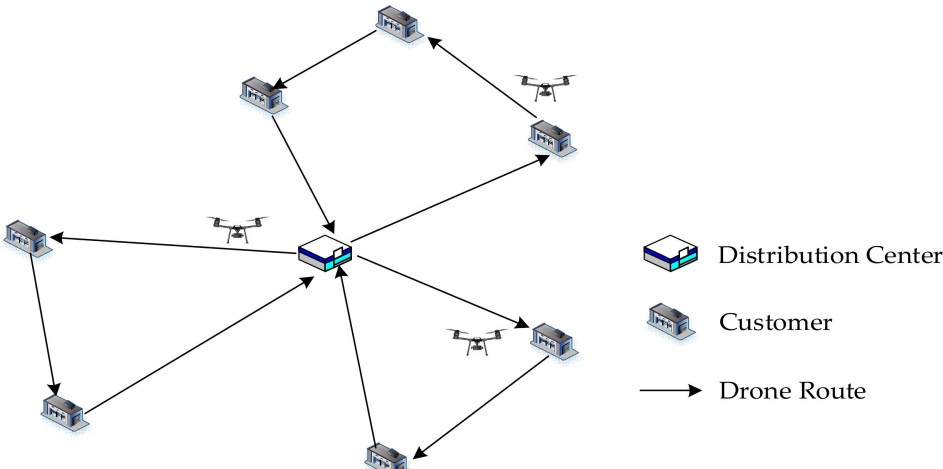

**Figure 1.** An illustration of the drone delivery routing problem.

The model makes the following assumptions: (1) the geographical location of the distribution center and customers is known and the distance between any two nodes is derived with Euclidean distance; (2) this paper considers medical supplies-face shields that are in great demand under major public health emergencies, other medical materials are not considered.

The primary decision of DRPTW studied in this paper is to arrange each drone route by considering the drone load capacity, the maximum continuous flight distance, the customer demand, and the service time windows. In other words, the order in which the drone visits customers is arranged under a series of constraints. Considering the timeliness of emergency logistics, the decision objective is to minimize the total flight time traveled by all drones.

*3.2. Model Formulation*

The following defines the set, index, parameters, and decision variables for DRPTW in Table 3.

**Table 3.** Notations and definitions for DRPTW.

| Notations | Definitions |
|---|---|
| $N$ | Set of all nodes |
| $C$ | Set of customers |
| $N_0 = \{0\} \cup C$ | Set of starting node and customers |
| $N_{n+1} = \{n+1\} \cup C$ | Set of ending node and customers |
| $R$ | Set of drones |
| $E$ | Set of available links |
| $r$ | Drone index, $r = 1, 2, \ldots, |R|$ |

**Table 3.** *Cont.*

| Notations | Definitions |
|---|---|
| $i, j, g$ | Node index, $i, j, g \in N$ |
| $d_{ij}$ | Distance of link $(i, j)$, $d_{ij} \leq d_{ig} + d_{gj}$; $\forall i, j, g \in N$ |
| $L$ | The maximum sustainable flight distance by drone $r$, $\forall r \in R$ |
| $v$ | The flight speed of drone $r$, $\forall r \in R$ |
| $q_i$ | Quantity of materials demanded by customer $i$, $\forall i \in C$ |
| $Q$ | Load capacity of drone $r$, $\forall r \in R$ |
| $S_{ir}$ | Service start time of customer $i$, $\forall r \in R$, $i \in C$ |
| $t_i$ | Service duration time of the customer $i$, $\forall i \in C$ |
| $[a_i, b_i]$ | Time windows of customer $i$, $\forall i \in C$ |
| $x_{ijr}$ | 1, If drone $r$ travels in link $(i,j)$, 0, else, $\forall (i, j) \in E$ |
| $Q_{ir}$ | The amount of materials delivered by drone $r$ to customer $i$, $\forall r \in R$, $i \in C$ |

We present the mathematical formulation for the DRPTW as follows:

$$\min \sum_{(i,j) \in E} \sum_{r \in R} \frac{d_{ij} x_{ijr}}{v} + \sum_{i \in C} t_i \tag{1}$$

Subject to:

$$\sum_{j \in N_{0+1}} \sum_{r \in R} x_{ijr} = 1, \forall i \in C \tag{2}$$

$$Q_{ir} \leq q_i, \forall i \in C, r \in R \tag{3}$$

$$\sum_{i \in N_0} \sum_{j \in N_{n+1}} Q_{ir} x_{ijr} \leq Q, r \in R \tag{4}$$

$$\sum_{i \in N_0} \sum_{j \in N_{n+1}} d_{ij} x_{ijr} \leq L, r \in R \tag{5}$$

$$\sum_{j \in N_{0+1}} x_{0jr} = 1, \forall r \in R \tag{6}$$

$$\sum_{i \in N_0} x_{ijr} - \sum_{g \in N_{0+1}} x_{jgr} = 0, \forall j \in C, r \in R \tag{7}$$

$$\sum_{i \in N_0} x_{in+1r} = 1, \forall r \in R \tag{8}$$

$$S_{ir} + t_{ij} + t_i - M(1 - x_{ijr}) \leq S_{jr}, \forall i \in N_0, \forall j \in N_{n+1}, \forall r \in R \tag{9}$$

$$a_i \leq s_{ir} \leq b_i, \forall i \in C, \forall r \in R \tag{10}$$

$$x_{ijr} \in \{0, 1\}, \forall (i, j) \in E, \forall r \in R \tag{11}$$

Equation (1) denotes the objective function, which minimizes the sum of the total flight time of all drones and the total service time of all customers. Constraints (2) and (3) assure that each customer can only be visited once by the drone. Constraint (4) guarantees that the drone loading must not exceed the drone capacity. Constraint (5) guarantees that the drone's cumulative flight distance cannot exceed the maximum sustainable flight distance. Constraint (6) denotes that drones must depart from the distribution center. Constraint (7) indicates the continuity of the drone route. Constraint (8) means that the drone must return to the distribution center after servicing. Constraint (9) expresses the continuity of the drone service customer time. Constraint (10) indicates the drone service time cannot violate the customer service time windows. Constraint (11) represents a binary constraint.

### 3.3. Set Covering Model

The integer linear programming problem involves numerous variables and parameters and confronts the problem of slow solving. The Dantzig–Wolfe decomposition (DW) can decompose a decomposable complex linear programming model into a linear programming model with simpler constraints and a number of smaller subprograms. Hence, we utilize DW to reconstruct the original problem and decompose it into a path-based master problem (MP) and a pricing subproblem (PS) based on an elementary shortest path problem with resource constraints (ESPPRC).

#### 3.3.1. Master Problem

If we obtain a set of feasible paths $\Omega$ satisfying constraints (2)–(11), our problem transforms into selecting several $p \in \Omega$ from the set of all feasible routes $\Omega$ to form a feasible solution, which minimizes the objective function.

The drone routes $p \in \Omega$ all start from a distribution center, serve a series of customers and then return to the distribution center. If a drone route satisfies the maximum drone capacity constraint, the maximum sustainable flight distance, and the corresponding service time windows, it is considered feasible. Let $C_p$ denote the time of route $p \in \Omega$. Let $a_{ip} = 1$ if route $p$ visits node $i$ and 0 otherwise. Let $b_{ijp} = 1$ if route $p$ travels $(i, j) \in E$ and 0 otherwise. Let $x_p = 1$ if route $p \in \Omega$ is selected in the final solution and 0 otherwise. The correlation variables are as follows:

$$C_p = \sum_{(i,j) \in E} b_{ijp} C_{ij}, \forall p \in \Omega \tag{12}$$

$$a_{ip} = \sum_{(i,j) \in E} b_{ijp}, \forall i \in C \tag{13}$$

The DRPTW can be described by the following set covering model known as the MP:

$$\min z = \sum_{p \in \Omega} C_p x_p \tag{14}$$

Subject to:

$$\sum_{p \in \Omega} a_{ip} x_p = 1, \forall i \in C \tag{15}$$

$$x_p \in \{0, 1\}, \forall p \in \Omega \tag{16}$$

The objective function (14) minimizes the total time. Constraint (15) means that each customer is visited only once. Constraint (16) denotes a binary constraint.

#### 3.3.2. Pricing Subproblem

Since a large number of feasible paths are contained in $\Omega$, it is difficult to directly solve the MP, therefore, the simplex method (SM) cannot be used to solve the current model. The CG has better performance for solving large-scale linear programming problems, so the CG will be adopted. The CG firstly constructs a linear relaxation of the MP, then finds a part of the feasible initial path to form the restricted master problem (RMP) to solve the model. The relaxation of the RMP model can be constructed as follows:

$$\min z = \sum_{p \in \Omega_1} C_p x_p \tag{17}$$

Subject to:

$$\sum_{p \in \Omega_1} a_{ip} x_p = 1, \forall i \in C \tag{18}$$

$$0 \leq x_p \leq 1, \forall p \in \Omega_1 \tag{19}$$

Let $\lambda_i (i \in C)$ be the dual variables of constraint (18), then the reduced cost for each feasible route is:

$$\overline{C_p} = C_p - \sum_{i \in C} a_{ip} \lambda_i, \text{ where } p \in \Omega_1 \tag{20}$$

According to the theory of SM, adding the column (path) of the reduced cost $\overline{C_p} < 0$ of the linear programming problem to the RMP for iteration can optimize the current optimal solution. The PS can be transformed into a search for the column (path) $\overline{C_p} < 0$. The PS model is described as follows:

$$\min \sum_{i \in N_0} \sum_{j \in N_{n+1}} (C_{ij} - \lambda_i) x_{ijr} \tag{21}$$

Subject to: Equations (2)–(11).

The PS can be known as an ESPPRC subproblem. The route sought by each PS must satisfy corresponding capacity constraints and time windows constraints. Add the solution to the RMP that solves the PS objective function value less than zero. The current result presents the optimal solution to the RMP relaxation when there is no column (path) that makes the PS objective function value less than zero.

## 4. Solution Method

In this paper, we embed the PA into a CG framework that solves the linear relaxation of the DRPTW.

### 4.1. Initial Solution

An initial set of feasible solutions is required to be construed in the CG algorithm. However, it is often impossible to list all feasible solutions. Therefore, it is necessary to apply definite approaches to generate an initial feasible route that covers all customer nodes and satisfies all constraints in a relatively short time period to form the initial solution space. Heuristic algorithms are exploited to generate initial solutions in the existing literature. To facilitate scheme design, this paper assumes that when the number of vehicles is sufficient, a separate path is created for each customer to constitute the initial route of the distribution center–customer–distribution center. The path obtained are as follows: Path 1: $0 \rightarrow 1 \rightarrow 0$; Path 2: $0 \rightarrow 2 \rightarrow 0$; Path 3: $0 \rightarrow 3 \rightarrow 0$......Path n: $0 \rightarrow n \rightarrow 0$, and set each path to $P_1$, $P_2$, $P_3...P_n$, to construct the initial RMP model.

### 4.2. Column Generation

The CG, a very efficient algorithm for solving large-scale linear optimization problems, was proposed by Danzig in 1960. Essentially, the CG algorithm operates through the idea of the SM. The overview of the CG algorithm is given in Figure 2. The algorithm first restricts the MP to a relatively small-scale problem, that is, the RMP. Dual values $\lambda_i (i \in C)$ obtained by addressing the RMP can be utilized to resolve the PS. Faced with the problem of minimizing the optimization, the column with the least reduced cost is added to the primal solution. The iteration cycle then continues until no more negative reduced costs are generated. In the case of a non-integer solution, the branch and bound procedure will be executed again.

### 4.3. Pulse Algorithm

Due to the large scale of the solution, solving the PS often consumes a large amount of time. In the past, many researchers used heuristic algorithms to solve ESPPRC. The PA can not only enhance the quality of the result, but also speed up the resolution of the PS to a certain limit. Therefore, it is applied to solve the DRPTW.

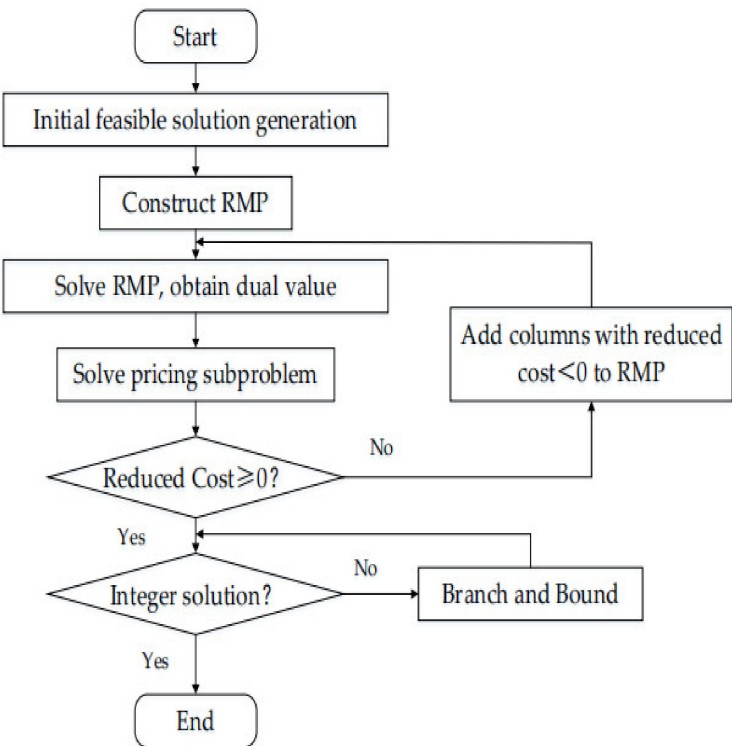

**Figure 2.** Overview of the CG algorithm.

The PA is a depth-first search schema, was proposed by Lozano in 2013 [26] and was subsequently extended in 2016 [27]. In terms of form, pulse propagation refers to a recursive exploration pattern in which a partial path is stretched until it reaches a terminal node or is discarded by a pruning strategy. The algorithm is generally divided into two stages: (1) Bound phase. In this phase, a bound matrix is established to store the lower bound of the remaining path cost at each point under a certain resource consumption. (2) Iterative expansion stage: Using a depth-first search schema, the pulse is propagated recursively until it reaches the terminal node or is pruned. Whenever a pulse reaches the end, update the current optimal solution until all possible attempts have been made to find the optimal solution to the problem. Three pruning strategies are followed in this process, namely infeasible pruning, bound pruning, and rollback pruning.

### 4.3.1. Bound Phase

The period pertains to the preliminary processing phase of the PA, which forms a bound matrix by determining the minimum cost to reach each node. The bound phase focuses on relaxing other resource constraints, assuming that the path has been consumed $\tau$, calculating the lowest cost of the point to the endpoint, thereby forming a bound matrix $B$. It is shown in Equation (22). Let $T$ be the upper time window at the distribution center and let $\Delta$ be a non-negative time step. Then we split the time window by the time step $\Delta$.

$$
B = \left\{
\begin{array}{ccccc}
c(1,0) & c(1,\Delta) & c(1,2\Delta) & \cdots & c(n,T) \\
c(2,0) & c(2,\Delta) & c(2,2\Delta) & \cdots & c(n,T) \\
c(3,0) & c(3,\Delta) & c(3,2\Delta) & \cdots & c(n,T) \\
\vdots & \vdots & \vdots & \ddots & \vdots \\
c(n,0) & c(n,\Delta) & c(n,2\Delta) & \cdots & c(n,T)
\end{array}
\right\} \tag{22}
$$

To avoid excessive pruning, the lower bound time should be less than the actual real-time when judging a node. For example, consider a problem with the upper time window $T = 100$ and the time step $\Delta = 10$. If the time consumption of a partial path to reach node 7 is 75, then the corresponding lower bound should be taken as $c(7,70)$

### 4.3.2. Iterative Expansion Stage

Although the PA does not generate labels like the label algorithm, each pulse transmitted from the source stores local path information $(i, q(p), t(p), r(p), p)$ from origin node 0 to the current node $i$. Here, $i$, the current node; $q(p)$, the cumulative capacity consumption; $t(p)$, the cumulative time consumption; $r(p)$, the cumulative reduced cost in the partial path $p$. Each pulse from the node $i$ is propagated to the outgoing node $j$, which will follow three pruning strategies for judgment. If the optimal solution cannot be acquired, the pulse will not continue to propagate the node $j$ and then try the outgoing node. Three pruning strategies are as follows:

1     Infeasible pruning

Whenever a partial path $p$ reaches a certain node $i$, the algorithm checks if the node satisfies various resources constraints (the time windows, the drone capacity). Regarding the time window $[a_i, b_i]$, if $t(p) < a_i$, the drone must wait until $a_i$ then to serve customer $i$. If $t(p) > b_i$, the drone violates the time window constraint, we will prune the partial path $p$ because node $i$ is visited after the latest time. Considering the drone capacity $Q$, if $q(p) > Q$, the demand of the customers visiting along the partial path $p$ exceeds the capacity of drone, thus, partial path $p$ is infeasible and pruned.

2.     Bound pruning

The acceleration strategy at this stage depends on the bound matrix, and the lowest reduced cost solved can play a delimiting role in the pulse search process, thus eliminating several bad paths. The current optimal reduced cost is $\bar{r}$ under the time that the current node $i$ is known to consume $t_p$. The current node $i$ ought to remove if $r_p + c(i, t_p) > \bar{r}$. Because no matter how backward from this node $i$, it will not get a better path than the current optimal solution.

3.     Rollback pruning

When the pulse propagates to node $j$, step back and re-evaluate whether a better solution is available if the node $i$ reaches the current node $j$ directly without passing through the previous node $k$. It can be seen from Figure 3 that the dotted line is more occupied than the solid line, so the dotted line is selected.

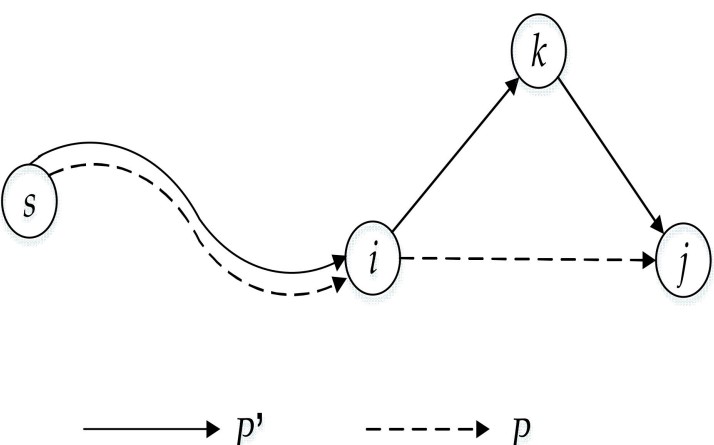

**Figure 3.** Dominance guidelines in rollback pruning strategy.

As shown in Figure 3, suppose that the path of the solid line is $p'$ and the path of the dashed line is $p$. If $p \subseteq p'$, $q(p) \leq q(p')$, $r(p) \leq r(p')$, $t(p) \leq t(p')$, at least one of these four conditions is satisfied. We call the path $p'$ is dominated by the path $p$. It is usually called the dominance guidelines.

## 5. Numerical Experiments

### 5.1. The Performance of Proposed Algorithm Test

The algorithm was coded in JAVA, IDE used was Eclipse, and the commercial solver used to solve the RMP problem was Cplex. All experiments were executed on the same computer. The computer configuration parameters were set as follows: Intel Core i5-7200U, 2.5 GHz main frequency, 4 GB memory, and Windows 10 operating system.

To verify the performance of the proposed model and algorithm in this paper, we selected 25 and 50 data nodes from the Solomon standard datasets C101, C102, and C103, respectively, for testing. Table 4 shows the comparison results of the branch and bound (BB), TS algorithm, and our algorithm. The "Data Sets" column indicates the data set to be tested; The "TC/s" column presents the current optimal time cost of the data set to be tested; The "RT/s" column presents the total program runtime.

**Table 4.** Comparison of test results of different algorithms.

| Data Sets | BB | | TS | | CG + PA | |
|---|---|---|---|---|---|---|
| | TC/s | RT/s | TC/s | RT/s | TC/s | RT/s |
| C101-25 | 10,858.05 | 0.922 | 10,881.45 | 0.59 | 10,848.75 | 0.906 |
| C102-25 | 10,858.05 | 0.92 | 10,881.45 | 0.62 | 10,830.15 | 2.078 |
| C103-25 | 10,858.05 | 0.959 | 10,881.45 | 0.618 | 10,830.15 | 16.283 |
| C101-50 | 20,807.55 | 3.98 | 20,845.8 | 1.75 | 20,800.85 | 3.284 |
| C102-50 | 20,807.55 | 3.42 | 20,845.8 | 1.8 | 20,792.25 | 9.31 |
| C103-50 | 20,807.55 | 3.47 | 20,845.8 | 1.793 | 20,792.25 | 49.864 |

As can be seen from Table 4, the algorithm proposed in this paper obtains the lowest time cost compared with the other two algorithms. Therefore, the algorithm proposed in this paper can guarantee the solution quality well. The solution efficiency of the TS algorithm is quite high, but it requires multiple solutions to obtain a satisfactory solution and is not very robust. Our algorithm has strong robustness in solving. The solution time of the proposed algorithm increases gradually as the case size increases, which is related to the case size and does not affect the quality of the solution. In summary, our proposed algorithm performs well in terms of solution quality and robustness of the solution.

### 5.2. Instance Verification

A distribution center providing medical face shields dispatch services to multiple closed communities in X City under COVID-19 is selected as a simulation case. Parameter setting primarily adopts a combination of reference actual data and simulation.

The specific calculation example is as follows: 10 closed communities need to be dispatched within time windows, numbered 1, 2, 3,..., 10, and 0 represents the distribution center. The quantity of requirements per community is distributed according to two face shields per person and the actual demand data is shown in Table 5. The specific coordinates from the distribution center to multiple closed communities are achieved through Baidu Maps. The coordinates of specific nodes in the network are shown in Table 6, so that the distance between any two nodes can be obtained according to the specific coordinates, in km. The parameters related to the drone are shown in Table 7. Assuming that the drones deliver materials at 15:00, the customer service time windows information is shown in Table 8. Each customer service time is about 30 s.

**Table 5.** Demand for face shields in 10 enclosed communities.

| Number | 1 | 2 | 3 | 4 | 5 | 6 | 7 | 8 | 9 | 10 |
|---|---|---|---|---|---|---|---|---|---|---|
| Demand/n | 2750 | 2000 | 1500 | 750 | 2500 | 3750 | 750 | 500 | 1750 | 2500 |

**Table 6.** Network node coordinates.

| Number | 0 | 1 | 2 | 3 | 4 | 5 | 6 | 7 | 8 | 9 | 10 |
|---|---|---|---|---|---|---|---|---|---|---|---|
| X Coordinate | 1.9 | 0 | 0.206 | 1.1 | 1.6 | 1.4 | 1.8 | 3.5 | 4.3 | 2 | 3.2 |
| Y Coordinate | 0.921 | 0.97 | 1.4 | 1.7 | 0 | 2.1 | 1.9 | 1.6 | 0.151 | 2.4 | 2.9 |

**Table 7.** Drone parameters.

| | Available Number | Capacity/kg | Flight Speed/km/h | Endurance/h |
|---|---|---|---|---|
| Parameters | 8 | 20 | 100 | 0.5 |

**Table 8.** Customer service time windows.

| Number | 0 | 1 | 2 | 3 | 4 | 5 | 6 | 7 | 8 | 9 | 10 |
|---|---|---|---|---|---|---|---|---|---|---|---|
| Start Window | 15:00 | 15:21 | 15:02 | 15:04 | 15:05 | 15:15 | 15:11 | 15:05 | 15:02 | 15:05 | 15:09 |
| End Window | 15:30 | 15:27 | 15:03 | 15:05 | 15:06 | 15:21 | 15:14 | 15:10 | 15:04 | 15:07 | 15:15 |

The continuous iterative optimization process of linear relaxation of MP and PS is shown in Figure 4. The optimal value of the objective function is 1051.14 s after 16 iterations. In the first and the second iteration, the value of the objective function has changed considerably. This is mainly attributable to the fact that the original routes are set to be assigned as separate routes for each customer node. After one iteration, the optimization of the objective function value is remarkable. The objective function value of the PS is continuously iterated and the path with the minimum objective function value of the PS is taken to add the MP to continuously optimize the objective function value of the MP. Total time for the entire simulation case run results in 0.262 s.

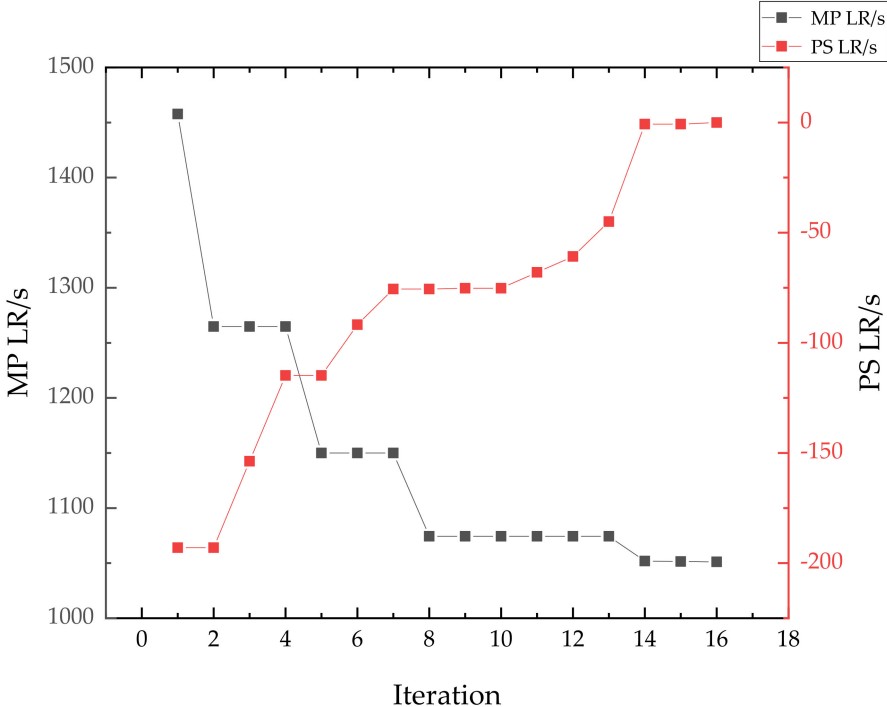

**Figure 4.** Iterative optimization process.

The specific delivery routes are shown in Figure 5. Each delivery route satisfies the drone endurance constraints and time window constraints. During the entire distribution

process, a total of 5 drones are used to deliver medical supplies. The drone delivery routes are as follows: route 1: $0 \to 8 \to 7 \to 10 \to 0$; route 2: $0 \to 2 \to 1 \to 0$; route 3: $0 \to 9 \to 5 \to 0$; route 4: $0 \to 4 \to 6 \to 0$; route 5: $0 \to 3 \to 0$. The distribution route map allows drones to be assigned to emergency logistics decision-makers for medical supply distribution in the shortest possible time.

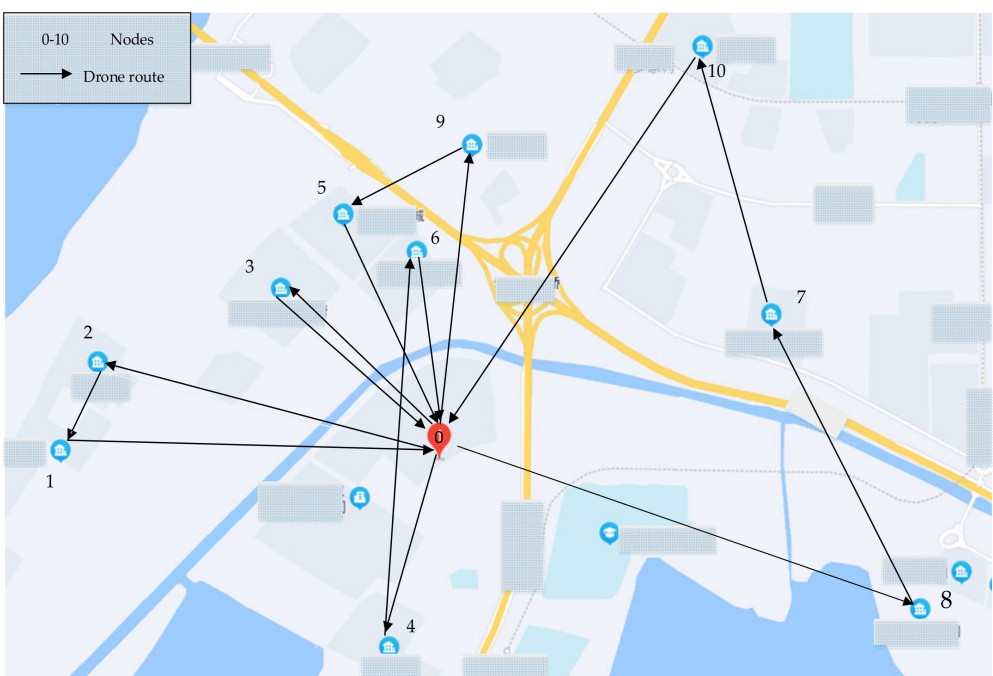

**Figure 5.** Route map of medical supplies delivery by drones in X city.

### 5.3. Comparison among Multiple Drones

The previous sections of this paper have concentrated on DRPTW, which indeed captures practical considerations under major public health emergencies. In this section, we perform a comparison of multiple drones materials delivery routes and time costs for a capacity of 20, 30, 40, and 50 kg, respectively. The analysis is performed based on a randomly generated data set.

A comparison of the test results of multiple drones is shown in Table 9. It is obvious that the drone with a capacity of 20 has the highest delivery time cost. With increased load capacity, drones delivery time costs are reduced by 16.7%. Likewise, the overall program runtime is significantly reduced. We can identify the iterative process of optimizing the time cost of different types of drones for materials delivery in Figure 6. It can be seen that the capacity of the drone has a significant impact on the delivery time cost of the drone.

**Table 9.** Comparison of test results of multiple drones.

| Capacity | N-Path | Route | MP LR/s | RT/s |
|:---:|:---:|:---:|:---:|:---:|
| 20 | 8 | $0 \to 3 \to 13 \to 18 \to 0$; $0 \to 8 \to 10 \to 0$; $0 \to 19 \to 15 \to 12 \to 0$; $0 \to 4 \to 1 \to 0$; $0 \to 16 \to 17 \to 6 \to 0$; $0 \to 11 \to 2 \to 0$; $0 \to 14 \to 20 \to 0$; $0 \to 9 \to 5 \to 7 \to 0$; | 10341.24 | 0.85 |
| 30 | 6 | $0 \to 9 \to 3 \to 13 \to 18 \to 0$; $0 \to 19 \to 5 \to 8 \to 2 \to 0$; $0 \to 14 \to 20 \to 0$; $0 \to 7 \to 4 \to 1 \to 0$; $0 \to 16 \to 15 \to 12 \to 0$; $0 \to 11 \to 17 \to 6 \to 10 \to 0$; | 9030.84 | 0.703 |

**Table 9.** *Cont.*

| Capacity | N-Path | Route | MP LR/s | RT/s |
|---|---|---|---|---|
| 40 | 5 | $0 \rightarrow 11 \rightarrow 17 \rightarrow 6 \rightarrow 0$; $0 \rightarrow 9 \rightarrow 15 \rightarrow 12 \rightarrow 0$; <br> $0 \rightarrow 14 \rightarrow 20 \rightarrow 3 \rightarrow 13 \rightarrow 18 \rightarrow 0$; <br> $0 \rightarrow 19 \rightarrow 5 \rightarrow 8 \rightarrow 2 \rightarrow 10 \rightarrow 0$; <br> $0 \rightarrow 16 \rightarrow 7 \rightarrow 4 \rightarrow 1 \rightarrow 0$; | 8662.56 | 0.689 |
| 50 | 5 | $0 \rightarrow 19 \rightarrow 0$; $0 \rightarrow 11 \rightarrow 17 \rightarrow 6 \rightarrow 0$; <br> $0 \rightarrow 14 \rightarrow 20 \rightarrow 3 \rightarrow 13 \rightarrow 18 \rightarrow 0$; <br> $0 \rightarrow 16 \rightarrow 5 \rightarrow 4 \rightarrow 7 \rightarrow 1 \rightarrow 0$; <br> $0 \rightarrow 9 \rightarrow 12 \rightarrow 15 \rightarrow 8 \rightarrow 2 \rightarrow 10 \rightarrow 0$; | 8606.04 | 0.657 |

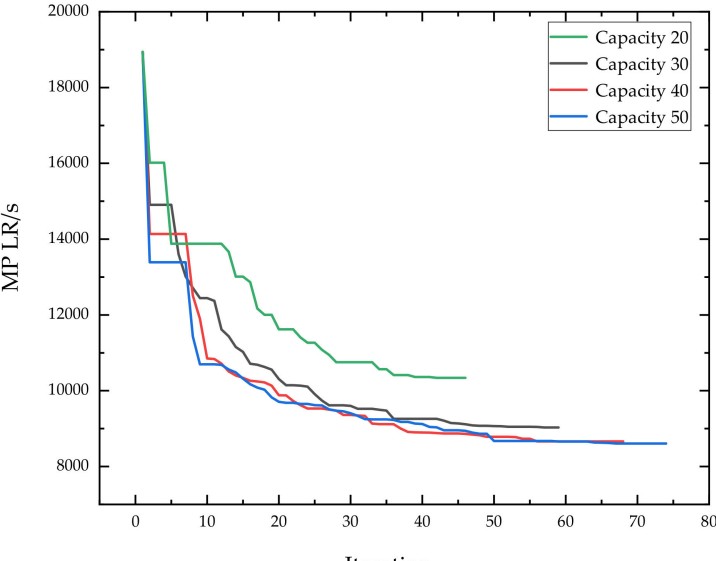

**Figure 6.** Comparison of the iterative optimization process of different types of drones.

Therefore, considering the actual materials distribution process, choosing drones with the right load capacity will not only effectively reduce time costs, but also reduce the number of drones dispatches.

## 6. Conclusions

In this paper, the following conclusions can be drawn through the discussion of the medical materials distribution drone routing problem under major public health emergencies.

To minimize the total delivery time, an optimization model that integrates road blockage policy, contactless delivery, customers' service time windows, and drone endurance is developed to reflect the actual supplies distribution under major public health emergencies. The resulting drones scheduling optimization solutions can effectively decrease the total delivery time for medical materials distribution. Embedding the PA into a CG framework can ensure the quality and robustness of the solutions in resolving. The established DRPTW model can effectively solve the single-depot homogeneous drone routing problem for traffic blockades and reduce the risk of personnel exposure under epidemic materials distribution. Depending on the actual volume of customer demand, drones of different capacities can be selected for delivery, which will improve the efficiency of delivery to a certain extent. This is an enrichment of the DRPTW study.

The research not only enriches the existing research related to vehicle routing problem models and algorithms under major public health emergencies, but also provides feasible optimized distribution solutions for emergency logistics decision-makers. Due to the limited flight range of drones, there are limitations in the range of delivery. In the future,

we should consider the joint distribution of multiple types of vehicles to achieve a more efficient distribution of supplies and pursuit more high-quality solutions.

**Author Contributions:** L.D. contributed to providing the core idea of this paper and provision of valuable comments. X.L. analyzed the data and wrote the manuscript. Y.G. and K.L. collected relevant data and information. All authors have read and agreed to the published version of the manuscript.

**Funding:** This research was funded by the National Natural Science Foundation of China, grant number 72042015; Foundation of Social Science and Humanity, China Ministry of Education, grant number 20YJC630018; Hubei Provincial Natural Science Foundation, grant number 2020CFB162.

**Institutional Review Board Statement:** Not applicable.

**Informed Consent Statement:** Not applicable.

**Data Availability Statement:** Not applicable.

**Acknowledgments:** We would like to appreciate editors' reviews and anonymous for their constructive reviews. Thanks to the authors of the references for their ideas.

**Conflicts of Interest:** The authors declare no conflict of interest.

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
