# Peer review of "Optimal Model and Algorithm of Medical Materials Delivery Drone Routing Problem under Major Public Health Emergencies"

_sustainability, doi:10.3390/su14084651_

Round 1

Reviewer 1 Report

  1. On page 8, the subchapter numbering is incorrect. There is 3.1.1, it should be 3.3.1 Master problem.
  2. On page 9, the subchapter numbering is incorrect. There is 3.1.2, it should be 3.3.2 Pricing Subproblem.

Author Response

Point 1: On page 8, the subchapter numbering is incorrect. There is 3.1.1, it should be 3.3.1 Master problem..

Response 1: We first appreciate very much this reviewer for his helpful comments on our manuscript. The numbering of subchapters on page 8 has been modified to 3.3.1 Master problem.

Point 2: On page 9, the subchapter numbering is incorrect. There is 3.1.2, it should be 3.3.2 Pricing Subproblem.

Response 2: Thanks for spotting this. The numbering of subchapters on page 9 has been modified to 3.3.2 Master problem.

Reviewer 2 Report

Dear authors,

First of all, thank you for the huge effort for elaborating such an interesting manuscript.

The article addresses the very important issue of reducing distribution risk and improving the efficiency of medical supplies in the context of major public health emergencies. The authors present the problem of drone routing with time windows. The authors illustrate the application of the proposed model and algorithm in practice and carry out a sensitivity analysis on drone capacity, which can affect the total distribution time. The paper is interesting and generally it deserves to be published with some revisions that are suggested below:

  1. In the "Conclusions" section, try to refer to previous research results of other authors and highlight the novelty of your results in comparison with previous research results.
  2. Please delete a blank page (page 18)

Author Response

Point 1: In the "Conclusions" section, try to refer to previous research results of other authors and highlight the novelty of your results in comparison with previous research results.

Response 1: Thanks to the reviewer for spotting this. In the Introduction section of the article, we give the innovative points of the research in this paper, which consist of three main points. Firstly, the study of medical supplies distribution under major public health emergencies is novel at present. Secondly, we established a drone routing model with time windows under major public health emergencies, which is an extension of the vehicle routing problem (Peng et al.2020). Drones distribution will achieve contactless delivery and be less affected by the road network. Finally, we avoided the employment of heuristic algorithms, and we adopt the column generation and pulse algorithm to solve the model, which largely ensures the quality and robustness of the understanding (Wang et al.2021; Zhang et al.2020).

1)Peng, B, Zhang, Y, Gajpal, Y, Chen, XD. A Memetic Algorithm for the Green Vehicle Routing Problem. Sustainability, 2020, 11(21).

2)Wang Y, Peng SG, Xu M. Emergency logistics network design based on space-time resource configuration. Knowledge-Based Syst. 2021, 223.

3) Zhang MinXia, Yan HongFan, Wu JiaYu, Zheng YuJun. Quarantine Vehicle Scheduling for Transferring High-Risk Individuals in Epidemic Areas. Int. J. Environ. Res. Public Health. 2020, 17(7).

Point 2: Please delete a blank page (page 18).

Response 2: Thanks for spotting this. We have removed the blank page on page 18.

Reviewer 3 Report

The authors in this paper are addressing an interesting problem which is the delivery of medical material in health emergency situations while showing a promising use for drones. The authors proposed and presented an algorithm that adopts a mixed column generation and pulse algorithm to solve the routing problem between the distribution centers (for medical supplies) and the customers.

However, in order to really show the feasibility and prove efficiency, the authors should provide a comparison with enough experiments between the proposed algorithm and some of the heuristic algorithm(s) and mathematical programming algorithm(s).

The contribution and the difference between the proposed algorithm and the other optimization algorithms discussed in the related work must be mentioned clearly in points.

Tables 4 and 5 are not clear, descriptive information below these tables explaining the presented results will be helpful.

There is an empty page (page 18)

Author Response

Point 1: However, in order to really show the feasibility and prove efficiency, the authors should provide a comparison with enough experiments between the proposed algorithm and some of the heuristic algorithm(s) and mathematical programming algorithm(s).

Response 1: Thanks to the reviewer for spotting this. Your suggestions are of great help to our article. Following your suggestion, we conducted a series of experiments in section 5.1 of the article. We compare the results of our algorithm with the branch and bound algorithm and the tabu search algorithm for the analysis of this paper. The results show that the proposed algorithm can guarantee the quality and robustness of the solution.

Point 2: The contribution and the difference between the proposed algorithm and the other optimization algorithms discussed in the related work must be mentioned clearly in points..

Response 2: Thanks for spotting this. In the Introduction section of the article, we give our contributions of the research in this paper, which consist of three main points. Firstly, the study of medical supplies distribution under major public health emergencies is novel at present. Secondly, we established a drone routing model with time windows under major public health emergencies, which is an extension of the vehicle routing problem (Peng et al.2020). Drones distribution will achieve contactless delivery and be less affected by the road network. Finally, we avoided the employment of heuristic algorithms, and we adopt the column generation and pulse algorithm to solve the model, which largely ensures the quality and robustness of the understanding (Wang Y et al.2021; Zhang MinXia et al.2020).

1)Peng, B, Zhang, Y, Gajpal, Y, Chen, XD. A Memetic Algorithm for the Green Vehicle Routing Problem. Sustainability, 2020, 11(21).

2)Wang Y, Peng SG, Xu M. Emergency logistics network design based on space-time resource configuration. Knowledge-Based Syst. 2021, 223.

3) Zhang MinXia, Yan HongFan, Wu JiaYu, Zheng YuJun. Quarantine Vehicle Scheduling for Transferring High-Risk Individuals in Epidemic Areas. Int. J. Environ. Res. Public Health. 2020, 17(7).

Point 3: Tables 4 and 5 are not clear, descriptive information below these tables explaining the presented results will be helpful.

Response 3: Thanks to the reviewer for spotting this. We provide some explanation of tables 4 and 5 in our manuscript to make them clearer.

Point 4: There is an empty page (page 18).

Response 4: Thanks for spotting this. We have removed the blank page on page 18.

Round 2

Reviewer 3 Report

the authors addressed the comments mentioned on their first submission.